# Correlation between Blood Lipid Level and Osteoporosis in Older Adults with Type 2 Diabetes Mellitus—A Retrospective Study Based on Inpatients in Beijing, China

**DOI:** 10.3390/biom13040616

**Published:** 2023-03-29

**Authors:** Xin Zhao, Jianbin Sun, Sixu Xin, Xiaomei Zhang

**Affiliations:** Department of Endocrinology, Peking University International Hospital, Beijing 100001, China

**Keywords:** type 2 diabetes mellitus, osteoporosis, low-density lipoprotein cholesterol, high-density lipoprotein cholesterol, bone mineral density

## Abstract

Objective: to analyze the association between blood lipid metabolism and osteoporosis (OP) in older adults with type 2 diabetes mellitus (T2DM). Methods: a total of 1158 older patients with T2DM treated by the Department of Endocrinology, Peking University International Hospital, were retrospectively analyzed, including 541 postmenopausal women and 617 men. Results: (1) Levels of low-density lipoprotein cholesterol (LDL-C) were significantly higher in the OP group, while levels of high-density lipoprotein cholesterol (HDL-C) were higher in the non-osteoporotic group (both *p* < 0.05). (2) Age, parathyroid hormone (PTH), total cholesterol (TC) and LDL-C were negatively linked to the patients’ bone mineral density (BMD) (all *p* < 0.05), while the body mass index (BMI), uric acid (UA) level, HDL-C level and glomerular filtration rate (eGFR) were positively related to their BMD (all *p* < 0.05). (3) In postmenopausal women, after adjustment for other indexes, raised LDL-C is an independent risk factor for OP (OR = 3.38, 95% CI 1.64, 6.98, *p* < 0.05) while raised HDL-C is protective (OR = 0.49, 95% CI 0.24, 0.96, *p* < 0.05). However, raised HDL-C was protective against OP (OR = 0.07, 95% CI 0.01, 0.53, *p* < 0.05). Conclusion: In older T2DM patients, the effect of blood lipid levels is related to sex. Our study conducted a detailed sex stratification. In addition to seeing the traditional risk factors of OP, such as age, sex, and BMI, we comprehensively analyzed the correlation between the blood glucose level, complications, and blood lipids with OP. HDL-C is a protective factor for OP in both men and women, while LDL-C independently predicts OP in postmenopausal women.

## 1. Introduction

Changes in lifestyle and demographics have led to an increased prevalence of type 2 diabetes mellitus (T2DM) and osteoporosis (OP), both characteristic of aging populations, and which frequently coexist [1]. Due to the large number of patients with these diseases, T2DM is often complicated with OP, leading to significant increases in the bone fragility of these patients. OP can lead to a high level of disability, which not only increases the economic burden on patients’ families but also seriously affects the patients’ quality of life. Di Somma et al. [2] observed that diabetes is a risk factor for OP, and that T2DM adversely affects both the bone mass and bone tissue structure of patients. Research shows that both T1DM and T2DM can increase the likelihood of osteoporotic fracture [3]. Relative to the normal population, diabetic patients have higher mortality after fracture and a higher probability of complications such as renal dysfunction and cardiovascular complications, leading to a poor prognosis [4]. Therefore, it is particularly urgent to accurately identify the abnormal bone mass of T2DM patients and predict the fracture risk.

The pathogenesis of osteoporosis has not been fully clarified, and there are many risk factors leading to its occurrence, including uncontrollable factors such as race, age, sex, a family history of fractures; and controllable factors such as weight, estrogen level, smoking, drinking, and exercise duration. The estrogen deficiency caused by menopause increases the production of osteoblasts and osteoclasts, increases the rate of bone turnover, and increases the apoptosis of osteoblasts and osteocytes [5]. The level of vitamin D in the elderly decreases, the proliferation and differentiation of osteoblasts is low, and bone formation is reduced. These factors act on different stages of osteogenesis or osteoclasis by raising the level of bone turnover, with bone absorption being greater than bone formation, and ultimately leading to bone loss. Increased bone resorption leads to porous cortical bone and increased bone fragility; the loss of bone mass causes diffuse fractures of bone trabecula, which become prone to fracture.

Patients with diabetes often have dyslipidemia, with above 75% of T2DM patients having dyslipidemia [6]. However, the potential relationship between the specific blood lipid components and OP is not clear, and the research results are quite controversial. Cardiovascular disease and OP share risk factors, such as age, sedentary lifestyle, smoking, menopause, and deficiencies in vitamin D [7,8]. Lessons from unloading, microgravity, and disuse teach us that gratuitous tissue is removed or reorganized while immobility and inflammation trigger muscle and bone marrow fatty infiltration and propagate degenerative diseases such as sarcopenia and osteoporosis [9]. Genetics and physiological factors may contribute to these connections. Physiological factors include those influencing atherosclerosis and bone remodeling, such as lipid oxidation, osteoprotegerin, sclerosing protein, metastatic breast cancer cells and fibroblast growth factor 23 (FGF-23) [10]. A study has shown the increased dickkopf-1 serum levels and decreased Trabecular Bone Score (TBS) values observed suggest diffuse bone damage in patients with advanced disease, as demonstrated by the concomitant presence of the “Late” pattern [11]. After menopause, the amount of estrogen drops sharply and inflammatory factors affect the activity of osteoclasts, leading to bone loss. Drugs used to treat osteoporosis and dyslipidemia, such as bisphosphonates and statins, also have effects on bone tissue and atherosclerotic plaques [12].

Because of the limitations of sample size, many results of the previous studies were controversial. The number of studies on hospitalized patients with T2DM in the Chinese population is very small, especially in elderly diabetes patients. There was no specific study on this population in previous studies. Therefore, we retrospectively analyzed the relationships between various blood lipid components and OP in older T2DM patients, so as to add a new direction for the discovery of OP risk factors in these patients and to provide evidence for preventing OP and fractures in such patients.

## 2. Research Methods

### 2.1. Participants

The data of 1158 T2DM patients over 50 years old who had been treated at the Department of Endocrinology, Peking University International Hospital, between January 2017 and August 2022 were retrospectively analyzed. The average age was 63.54 ± 7.82 years (50–80 years) and the patients included 541 postmenopausal women and 617 men. All participants met the T2DM diagnostic criteria [13] defined by the World Health Organization (WHO) in 1999. Exclusion criteria: (1) non-T2DM patients, including patients with T1DM, or other specific type of DM; (2) non-physiological postmenopausal women; (3) patients on long-term medication with drugs affecting bone metabolism; (4) patients with a history of bone malignancies, either primary or secondary; (5) patients treated with OP drugs, such as estrogen bisphosphonates and active vitamin D. This study was approved by the Ethics Committee of the Peking University International Hospital. The study was conducted in accordance with the Declaration of Helsinki, and approved by the bioethics committee of Peking University International Hospital. The date of ethics is 22 July 2022 and the number is 2022-KY-0030-01).

### 2.2. Clinical Information

Data were obtained from the patients’ medical records and included: age; weight; height; systolic blood pressure (SBP); diastolic blood pressure (DBP); T2DM duration; complications including diabetic kidney disease (DKD), diabetic retinopathy (DR), and diabetic neuropathy (DN). The body mass index (BMI) was determined as weight (kg)/height^2^ (m^2^).

### 2.3. Laboratory Measurement

Ten milliliters of fasting venous blood were collected from each patient. Fasting blood glucose (FBG), uric acid (UA), glycosylated hemoglobin (HbA1c), calcium (Ca), serum creatinine (sCr), triglyceride (TG), total cholesterol (TC), low-density lipoprotein cholesterol (LDL-C), high-density lipoprotein cholesterol (HDL-C), osteocalcin (OC), parathyroid hormone (PTH), procollagen 1 N-terminal propeptide (P1NP), β-crosslaps peptide (β-CTx), and 25-hydroxyvitamin D [25(OH)D] levels were measured.

The FBG, sCr, UA, Ca, TC, TG, LDL-C, HDL-C, and PTH concentrations were determined with enzymatic methods. OC, β-CTx, P1NP, and 25(OH)D were measured using electrochemiluminescence assays, while HbA1c was measured with high-performance liquid chromatography.

### 2.4. Bone Mineral Density (BMD) Determination

A Discovery QDR SERIES dual-energy X-ray bone density instrument (Hologic, Bedford, MA, USA) was used to determine the bilateral hip and lumbar vertebral (L1–4) BMD. Quality control assessments were conducted daily on the instrument. The coefficient of variation was 1.0%. *T*-scores were determined automatically with the instrument’s software, using the Asian population as reference. OP is defined by the WHO as a *T*-score of ≤ −2.5 standard deviation (SD) at any site, with osteopenia defined as −1.0 ≥ *T*-score ≥ −2.5. The *T*-scores of the patients were used to establish three patient groups, namely, the normal, osteopenia, and OP groups.

### 2.5. Statistical Analysis

All data were analyzed using SPSS 25.0 (SPSS Corporation, Chicago, IL, USA) Measurement data were tested for normality of distribution and homogeneity of variance. Normally distributed data were expressed as mean standard deviation (x s) and non-normally distributed data as median and interquartile spacing. Analysis of variance was used for the comparison of data with normal distribution and homogeneity of variance. Categorical data were expressed as numbers and percentages and analyzed with chi-square tests. Correlation analyses included both Pearson and Spearman correlations. Logistic regression method was used for univariate and multivariate analysis of the factors, and OR and its 95% confidence interval (CI) were calculated. *p* < 0.05 was considered statistically significant.

## 3. Results

### 3.1. Patient Features and Measurements in the Different Groups

The OP group was characterized by having more female members of an older age and a lower BMI, differing significantly from normal and osteopenia patients (*p* < 0.05, respectively). GFR was also reduced in the OP group (*p* < 0.05) and was highest in the normal group (*p* < 0.05), while the UA was significantly lower in the OP patients and highest in normal BMD patients (*p* < 0.05). Furthermore, LDL-C concentrations were markedly higher in both the OP and osteopenia patients, while HDL-C was markedly higher in the non-osteoporotic group (*p* < 0.05, respectively). The levels of β-CTx, OC, and P1NP were lowest in the non-osteoporotic group and highest in the OP group (*p* < 0.05, respectively) while normal patients had higher levels of 25 (OH) D, in contrast to OP patients, where the levels were significantly lower (*p* < 0.05, respectively). We further conducted grouping analysis based on sex, and the results showed that in elderly male patients, the OP group was characterized by having lower BMI, lower HDL-C (*p* < 0.05, respectively), and the levels of β-CTx, OC, and P1NP were lowest in the non-osteoporotic group, and highest in the OP group (*p* < 0.05, respectively). In female patients, the OP group was characterized by having older aged patients, lower BMI and HDL-C, which differed significantly from normal and osteopenia patients (*p* < 0.05, respectively). GFR was also reduced in the OP group (*p* < 0.05) and highest in the normal group (*p* < 0.05). The levels of β-CTx and OC were lowest in the non-osteoporotic group and highest in the OP group (*p* < 0.05, respectively). Both in male and female patients, the history of fractures in OP group was highest (*p* < 0.05) and the DKD portion in the OP group was highest (*p* < 0.05) (shown in Table 1 and Table 2).

### 3.2. Correlations between BMD and Blood Lipid Levels in T2DM Patients

In male patients, the hip BMD value was negatively associated with age (*r* = −0.15, *p* < 0.05) but positively associated with BMI (*r* = 0.30, *p* < 0.05), and the BMD value of the lumbar spine was positively associated with eGFR (*r* = 0.12, *p* < 0.05). Also, the BMD value of the hip and lumbar spine were positively associated with HDL-C (*r* = 0.34; *r* = 0.18, *p* < 0.05, respectively).

In female patients, the BMD value of the hip and lumbar spine were negatively associated with age (*r* = −0.39; *r* = −0.25; *p* < 0.05, respectively), but positively associated with BMI (*r* = 0.13; *r* = 0.21; *p* < 0.05, respectively), and the BMD value of the hip was positively associated with eGFR (*r* = 0.24, *p* < 0.05). Also, the BMD value of the hip and lumbar spine were positively associated with HDL-C (*r* = 0.48; *r* = 0.31; *p* < 0.05, respectively) (shown in Table 3).

### 3.3. Logistic Regression of Blood Lipid Levels and OP in Male and Female Patients

The blood lipid indicators were divided into three levels, according to the third percentile of the blood lipid level of each component in the subjects. Logistic regression was conducted using OP as the dependent variable and statistically significant indices as independent variables. After adjustment for age, BMI, duration of diabetes, blood pressure, blood glucose, calcium, and complications of diabetes and eGFR, in postmenopausal women, raised LDL-C was observed to be an independent risk factor for OP (OR = 3.38, 95% CI 1.64, 6.98, *p* < 0.05), while raised HDL-C was protective against OP (OR = 0.49, 95% CI 0.24, 0.96, *p* < 0.05). No independent association between LDL-C and OP was seen for male patients after factor adjustment, while high levels of HDL-C remained protective (OR = 0.07, 95% CI 0.01, 0.53, *p* < 0.05) (shown in Table 4).

## 4. Discussion

Although there was much investigation into an association between T2DM and OP, the issue remains controversial. Even T2DM patients with normal or even high BMD have an increased risk of fractures [14]. This suggests that diabetes itself may cause changes in the bone characteristics, which may include abnormalities in the bone materials and macro- and micro-structures, resulting in higher fracture risk. However, fractures resulting from OP can be prevented. It is thus necessary to identify OP risk factors in T2DM patients so that early treatment can be instigated. Our study conducted a detailed sex stratification, where we comprehensively analyzed the correlation between the blood glucose level, complications, and blood lipids, with OP. The results showed that HDL-C is a protective factor for OP in both men and women, while LDL-C independently predicts OP in postmenopausal women.

The association between blood lipids and OP risk can be explained as follows: (1) Similar cell sources: adipocytes, osteoblasts, and vascular endothelial cells all differentiate from precursor cells and bone marrow stromal cells, and adipocytes and osteoblasts in the body can transform into each other on occasion. The number and volume of adipocytes in the bone marrow are sensitive to changes in the blood lipid levels. This leads to an increased pressure in the bone marrow cavity, resulting in a reduction in blood flow, which may eventually lead to the ischemic necrosis of bone cells and the loss of bone mass. (2) Involvement of inflammatory factors: it is known that many pro-inflammatory factors and adipokines are involved in the development of OP. Atherosclerosis promotes the release of these factors, which, in turn, affects bone metabolism and promotes further reductions in the BMD [15]. Peroxisome proliferator-activated receptor (PPARγ) is a major transcriptional regulator in adipose tissue and, specifically, promotes the differentiation of adipocytes. Bone marrow stem cells are more likely to differentiate into adipocytes but less into osteoblasts through PPARγ. PPARγ has a natural ligand, namely, oxidized LDL-C, and interaction between the two promotes the adipocyte differentiation pathway, thus reducing the osteoblast differentiation and, consequently, the bone density. (3) Influence on key signaling pathways: a disordered lipid metabolism also affects the osteoblast differentiation through interference with the key signaling pathways [16]. Fat cells can secrete cytokines, which can also regulate the state of bone. It was found that the higher the proportion of fat in bone marrow, the lower the bone trabecular density. Leptin is associated with fat and is known to be involved in bone metabolism. Leptin influences both the central and peripheral nervous systems, affecting bone metabolic processes [17]. However, the issue is controversial and there is no consensus.

Cholesterol and its metabolites can adversely affect osteoblasts. A meta-analysis of ten articles related to lipid profiles and postmenopausal osteoporosis found that serum HDL and TC levels tend to be higher in postmenopausal OP patients, and thus may be potentially useful indicators for OP in these patients [18]. A further study on 1240 hospitalized patients over the age of 65 explored and discussed the relationships between OP and cardiovascular disease in the elderly. It was found that both LDL-C and TC were linked to OP risk in the elderly patients. On the other hand, HDL-C and body weight were observed to protect against OP in the elderly [19]. Brownhil [20] concluded that TC was positively associated with the total BMD. Here, a negative relationship was observed between TC and hip and lumbar BMD, suggesting that raised TC is a risk factor for OP; however, this did not hold up after an adjustment for age, diabetes duration, BMI and blood glucose. Stratification based on sex indicated that for the postmenopausal women, moderate and high TC levels were independent risk factors for OP, while for elderly male patients with T2DM, the TC level was not predictive of OP risk after adjustment for age, diabetes duration, BMI, and blood glucose. Thus, it appears that the relationship between TC and OP differs according to sex.

TG levels in postmenopausal women tend to be above normal and this, together with the negative relationship between TG and BMD, suggests that TG may mediate OP that results from estrogen reduction [21]. Our analysis of hospitalized patients with T2DM showed no association between TG and BMD values of different sites (hip and lumbar vertebrae), irrespective of the sex of the patient. It is thus apparent that TG levels are not predictive for OP in hospitalized patients with T2DM.

A study from the National Health and Nutrition Examination Survey (NHANES) found that following adjustment for confounders, LDL-C was negatively linked to lumbar BMD. This was statistically valid for both men and women, shown by subgroup analyses after sex stratification. Further subgroup analyses after stratification by age indicated negative associations in the age group of 30–49 years. In the subgroup analysis stratified for BMI, there was an inverse correlation between BMD and LDL-C in overweight people (25 ≤ BMI < 30) [22]. A further NHANES study found a negative association between BMD and LDL-C. Reductions in LDL-C levels induced with statins was found to increase BMD, indicating protective effects [23]. Poli [24] found that increased LDL-C in postmenopausal women raised the OP risk. A meta-analysis of 12 studies (12,395 subjects) found no significant differences in the LDL-C levels in postmenopausal women with osteopenia [25]. Here, we found that LDL-C levels were markedly higher in OP patients when compared with both the normal and osteopenia groups and, as shown with a logistic regression analysis, independently predicted OP risk (OR = 2.28, 95% CI 1.39, 3.72, *p* < 0.05) after adjustment for various factors, including age and BMI. After a stratification by sex and adjustment for the above confounders, raised LDL-C remained an OP risk factor (OR = 3.38, 95% CI 1.64, 6.98, *p* < 0.05). However, this did not apply to older men, indicating that clinicians should consider the sex of the patient when assessing LDL-C and OP risk.

There is some controversy as to the relationship between HDL-C and OP. While an early study found no significant link between HDL-C and BMD values of the lumbar vertebrae and femur [26], a large-sample analysis study in China reported the risk factors for abnormal bone masses in T2DM might be female gender, advanced age, long duration of T2DM, low BMI, high levels of HDL-C, and diabetic microangiopathy [27]. A further study found that HDL-C and BMD were inversely correlated [28]. Martineau [29] found that in scavenger B1 receptor gene (Scarb1) knockout mice, the content of oxidized sterol from HDL in the liver decreased while the oxidized sterol content in the periphery increased, accompanied by an increase in the volume of trabecular bone but no significant alteration in the bone cortex, although the osteogenic surface area increased. Here, it was found that HDL-C was higher in T2DM patients with normal bone density when compared with those with OP or osteopenia (*p* < 0.05). Further regression analysis indicated that elevated HDL-C protected against OP after confounder adjustment (OR = 0.28, 95% CI 0.13, 0.63, *p* < 0.05), while further stratification by sex found that high HDL-C was protective in both postmenopausal women and older men (OR = 0.49, 95% CI 0.24, 0.96; OR = 0.07, 95% CI 0.01, 0.53, *p* < 0.05).

Due to the limitations of the sample size, many results of previous studies are controversial. The number of studies on hospitalized patients with T2DM in the Chinese population is very small, and the correlation between blood lipid level and OP is not paid much attention, especially in elderly diabetes patients. There was no specific study on this population in previous studies. Our study conducted a detailed gender stratification. In addition to seeing the traditional risk factors of OP, such as age, sex, BMI, we comprehensively analyzed the correlation between the blood glucose level, complications, and blood lipids with OP.

There are still a few limitations to this study. First, the participants were T2DM patients hospitalized during a certain time in one institution, which runs the risk of creating a sampling error caused by Berkson’s bias, and thus, the study may not represent all T2DM patients. Future studies should conduct multi-center research to eliminate the bias caused by different levels of hospitals, expand the population and sample content, and minimize the sampling error, so that the conclusion can be generalized to all patients with T2DM. Secondly, this is a retrospective study, therefore, some data, such as the medication of T2DM patients, are lacking. In future prospective studies, we will include more comprehensive indicators to reduce research bias. Thirdly, due to the limited time and funds available, no animal studies were conducted. Future studies should address this issue to elucidate the specific mechanisms responsible for the association between diabetes and OP. Finally, at present, research on the influence of lipid-lowering drugs such as statins on BMD is limited. Future research should be expanded to analyze the influence of blood-lipid components on BMD and bone metabolism in T2DM patients. T2DM and dysfunctional bone metabolism are both complex metabolic disorders, with numerous overlaps and interactions. At present, the exact pathogenesis of OP-related blood lipid components in T2DM patients cannot be fully explained, and clinical research and basic research are still needed to clarify its pathogenesis.

## 5. Conclusions

To sum up, this clinical study showed that age, female sex, and high LDL-C levels can increase the likelihood of OP in T2DM patients, while high BMI and high HDL-C levels are protective against OP. HDL-C is a protective factor for OP in both men and women, while LDL-C independently predicts OP in postmenopausal women. The results of this study provided clinical evidence for the occurrence of OP in elderly T2DM patients. Thus, clinicians should be aware of the importance of patients’ blood lipid levels and take necessary interventional measures for their control. In T2DM patients with dyslipidemia, especially LDL-C and HDL-C, we should pay attention to the bone health of such people and give drug intervention in advance.

## Figures and Tables

**Table 1 biomolecules-13-00616-t001:** Comparison of general characteristics, biochemical indexes, BMD and bone metabolism markers among the three groups.

Index	Non-Osteoporotic Group (*n* = 272)	Osteopenia Group (*n* = 536)	Osteoporosis Group (*n* = 350)	F(*X*^2^)	*p*
Male	167 (61.40%)	322 (60.07%)	128 (36.57%) ^a,b^	56.39	<0.05
Age (year)	61.40 ± 6.61	62.70 ± 7.71	66.39 ± 7.96 ^a,b^	39.20	<0.05
BMI (kg/m^2^)	26.50 ± 3.43	25.48 ± 3.16 ^a^	24.40 ± 3.22 ^a,b^	30.28	<0.05
Diabetes duration(year)	11.38 ± 8.69	12.13 ± 7.78	12.02 ± 8.38	0.79	0.45
WHR	0.95 ± 0.07	0.95 ± 0.07	0.94 ± 0.07	2.29	0.10
SBP (mmHg)	134.23 ± 17.72	133.12 ± 17.27	134.26 ± 18.95	0.58	0.56
DBP (mmHg)	78.90 ± 10.49	77.82 ± 10.33	76.86 ± 10.72	2.91	0.05
HbA1c (%)	8.57 ± 1.99	8.68 ± 1.92	8.50 ± 1.81	0.84	0.43
FBG (mmol/L)	8.60 ± 3.40	8.81 ± 3.40	8.61 ± 3.19	0.52	0.59
TC (mmol/L)	4.15 ± 1.09	4.14 ± 1.09	4.30 ± 1.18	2.32	0.10
TG (mmol/L)	1.72 ± 1.09	1.83 ± 1.29	1.70 ± 1.14	1.48	0.23
LDL-C (mmol/L)	2.42 ± 0.83	2.38 ± 0.85	2.54 ± 0.94 ^a,b^	3.89	<0.05
HDL-C (mmol/L)	1.15 ± 0.33	1.04 ± 0.26 ^a^	0.91 ± 0.35 ^a,b^	44.59	<0.05
UA (mmol/L)	339.40 ± 86.24	343.04 ± 87.52	322.18 ± 85.63 ^a,b^	6.26	<0.05
eGFR (mL/min/1.73^2^)	92.26 ± 15.42	89.94 ± 18.26	87.65 ± 18.94 ^a^	4.99	<0.05
Ca (mmol/L)	2.29 ± 0.09	2.30 ± 0.11	2.30 ± 0.10	1.20	0.30
PTH (pg/mL)	34.31 ± 13.01	35.64 ± 15.98	38.44 ± 17.02 ^a,b^	5.56	<0.05
Lumbar BMD (g/cm^2^)	1.11 ± 0.16	0.95 ± 0.11 ^a^	0.79 ± 0.14 ^a,b^	429.44	<0.05
Hip BMD (g/cm^2^)	0.87 ± 0.09	0.71 ± 0.08 ^a^	0.59 ± 0.10 ^a,b^	781.02	<0.05
OC (ng/mL)	11.03 ± 4.08	12.56 ± 6.19 ^a^	14.07 ± 7.56 ^a,b^	17.50	<0.05
β-CTx (ng/mL)	0.32 ± 0.18	0.40 ± 0.36 ^a^	0.43 ± 0.29 ^a^	10.49	<0.05
P1NP (ng/mL)	35.89 ± 17.20	39.95 ± 21.06 ^a^	44.31 ± 22.68 ^a.b^	12.26	<0.05
25(OH)D (ng/mL)	19.18 ± 8.64	17.78 ± 8.67 ^a^	17.13 ± 7.29 ^a^	4.48	<0.05
History of fractures	22 (8.09%)	61 (11.38%)	54 (15.43%)	14.33	<0.05
DKD	78 (28.68%)	198 (36.94%) ^a^	148 (42.29%) ^a,b^	10.98	<0.05
DR	90 (33.09%)	178 (33.21%)	121 (34.57%)	1.32	0.78
DN	86 (31.62%)	176 (32.84%)	108 (30.86%)	3.21	0.65

^a^: *p* < 0.05 compared with the non-osteoporotic group; ^b^: *p* < 0.05 compared with the osteopenia group. Abbreviations: SBP: systolic blood pressure; DBP: diastolic blood pressure; BMI: body mass index; FBG: fasting blood glucose; HbA1c: glycosylated hemoglobin; UA: uric acid; Ca: calcium; TC: total cholesterol, TG: triglycerid; LDL-C: low-density lipoprotein cholesterol; HDL-C: high-density lipoprotein cholesterol; PTH: parathyroid hormone; OC: osteocalcin; β-CTx: β-crosslaps peptide; P1NP: procollagen 1 N-terminal propeptide; 25(OH)D: 25-hydroxyvitamin D; eGFR: glomerular filtration rate; BMD: bone mineral density; DKD: diabetic kidney disease; DR: diabetic retinopathy; DN: diabetic neuropathy.

**Table 2 biomolecules-13-00616-t002:** Comparison of general characteristics, biochemical indexes, BMD and bone metabolism markers grouping analysis based on sex.

Index	Male					Female				
	Non-Osteoporotic Group (*n* = 167)	Osteopenia Group (*n* = 322)	Osteoporosis Group (*n* = 128)	F(*X^2^*)	*p*	Non-Osteoporotic Group (*n* = 105)	Osteopenia Group (*n* = 214)	Osteoporosis Group (*n* = 222)	F(*X^2^*)	*p*
Age (year)	62.04 ± 6.62	62.58 ± 7.90	63.43 ± 8.20	1.20	0.30	60.37 ± 6.51	62.87 ± 7.44 ^a^	68.09 ± 7.32 ^a,b^	50.11	<0.05
BMI (kg/m^2^)	26.51 ± 3.20	25.50 ± 3.16 ^a^	24.06 ± 2.75 ^a,b^	21.52	<0.05	26.49 ± 3.80	25.47 ± 3.17 ^a^	24.60 ± 3.47 ^a,b^	10.39	<0.05
Diabetes duration (year)	11.76 ± 8.33	12.38 ± 7.70	10.85 ± 8.45	1.69	0.19	10.77 ± 9.24	11.47 ± 7.89	12.69 ± 8.28	2.00	0.13
WHR	0.96 ± 0.07	0.96 ± 0.07	0.95 ± 0.06	1.40	0.25	0.93 ± 0.07	0.93 ± 0.07	0.93 ± 0.07	0.03	0.97
SBP (mmHg)	133.37 ± 17.40	133.36 ± 17.76	132.84 ± 17.91	0.04	0.96	135.61 ± 18.21	132.75 ± 16.53	135.08 ± 19.51	1.26	0.28
DBP (mmHg)	78.86 ± 10.70	78.86 ± 10.54	78.67 ± 10.83	0.02	0.98	78.97 ± 10.21	76.26 ± 9.81	75.81 ± 10.54 ^a^	3.63	<0.05
HbA1c (%)	8.34 ± 1.90	8.59 ± 1.92	8.37 ± 1.78	1.13	0.33	8.93 ± 2.10	8.81 ± 1.91	8.58 ± 1.83	1.30	0.27
FBG (mmol/L)	8.61 ± 3.69	8.84 ± 3.42	8.52 ± 3.39	0.47	0.62	8.59 ± 2.86	8.77 ± 3.37	8.67 ± 3.09	0.12	0.88
TC (mmol/L)	3.94 ± 0.99	3.98 ± 1.02	4.00 ± 1.27	0.16	0.85	4.48 ± 1.15	4.38 ± 1.14	4.46 ± 1.09	0.45	0.64
TG (mmol/L)	1.60 ± 0.91	1.73 ± 1.16	1.56 ± 1.07	1.36	0.26	1.91 ± 1.32	1.98 ± 1.44	1.77 ± 1.18	1.37	0.26
LDL-C (mmol/L)	2.31 ± 0.79	2.30 ± 0.83	2.39 ± 0.98	0.54	0.59	2.58 ± 0.87	2.50 ± 0.86	2.63 ± 0.91	1.25	0.29
HDL-C (mmol/L)	1.14 ± 0.35	1.03 ± 0.27 ^a^	0.95 ± 0.28 ^a,b^	14.97	<0.05	1.17 ± 0.30	1.07 ± 0.25	0.90 ± 0.38 ^a,b^	30.48	<0.05
UA (mmol/L)	343.92 ± 81.99	352.98 ± 85.96	332.31 ± 85.98	2.73	0.07	332.20 ± 92.71	327.74 ± 87.91	316.41 ± 85.09	1.45	0.24
eGFR (mL/min/1.73^2^)	90.96 ± 15.21	90.04 ± 18.46	90.61 ± 18.05	0.16	0.85	94.39 ± 15.61	89.80 ± 17.98	85.96 ± 19.27 ^a^	7.65	<0.05
Ca (mmol/L)	2.28 ± 0.09	2.30 ± 0.10	2.30 ± 0.10	1.46	0.23	2.29 ± 0.10	2.30 ± 0.12	2.30 ± 0.09	0.06	0.94
PTH (pg/mL)	33.69 ± 11.82	35.53 ± 17.04	37.18 ± 16.92	1.73	0.18	35.35 ± 14.78	35.81 ± 14.25	39.17 ± 17.08	3.14	<0.05
Lumbar BMD (g/cm^2^)	1.13 ± 0.16	0.98 ± 0.11 ^a^	0.84 ± 0.14 ^a,b^	185.21	<0.05	1.07 ± 0.16	0.90 ± 0.10	0.77 ± 0.13 ^a,b^	211.83	<0.05
Hip BMD (g/cm^2^)	0.90 ± 0.09	0.73 ± 0.07 ^a^	0.63 ± 0.09 ^a,b^	432.49	<0.05	0.83 ± 0.09	0.68 ± 0.08	0.56 ± 0.10 ^a,b^	332.91	<0.05
OC (ng/mL)	10.34 ± 3.65	11.34 ± 5.75	12.35 ± 4.94 ^a^	5.49	<0.05	12.15 ± 4.49	14.41 ± 6.38 ^a^	15.09 ± 8.60 ^a^	5.91	<0.05
β-CTx (ng/mL)	0.31 ± 0.17	0.34 ± 0.20	0.43 ± 0.30 ^a,b^	10.03	<0.05	0.34 ± 0.19	0.47 ± 0.50 ^a^	0.48 ± 0.28 ^a^	4.60	<0.05
P1NP (ng/mL)	32.62 ± 12.33	36.25 ± 19.44	40.97 ± 21.46 ^a,b^	7.31	<0.05	41.17 ± 22.08	45.55 ± 22.19	46.28 ± 23.20	1.83	0.16
25(OH)D (ng/mL)	20.14 ± 8.00	18.83 ± 8.15 ^a^	18.43 ± 8.15	1.72	0.18	17.57 ± 9.43	16.20 ± 8.23	16.36 ± 6.64	1.05	0.35
History of fractures	9 (5.39%)	20 (6.21%)	19 (14.84%) ^a,b^	32.65	<0.05	13 (12.38%)	41 (19.16%)	35 (15.77%) ^a,b^	8.79	<0.05
DKD	30 (17.96%)	97 (30.12%) ^a^	39 (30.47%) ^a^	8.78	<0.05	35 (33.33%)	101 (47.20%)	109 (49.10%) ^a^	7.86	<0.05
DR	37 (22.16%)	79 (24.53%)	23 (17.97%)	2.01	0.63	53 (50.47%)	99 (46.26%)	98 (44.14%)	4.01	0.34
DN	39 (23.35%)	81 (25.16%)	31 (24.22%)	1.98	0.76	47 (44.76%)	95 (44.39%)	77 (34.68%)	3.09	0.45

^a^: *p* < 0.05 compared with the non-osteoporotic group; ^b^: *p* < 0.05 compared with the osteopenia group. Abbreviations: SBP: systolic blood pressure; DBP: diastolic blood pressure; BMI: body mass index; FBG: fasting blood glucose; HbA1c: glycosylated hemoglobin; UA: uric acid; Ca: calcium; TC: total cholesterol, TG: triglycerid; LDL-C: low-density lipoprotein cholesterol; HDL-C: high-density lipoprotein cholesterol; PTH: parathyroid hormone; OC: osteocalcin; β-CTx: β-crosslaps peptide; P1NP: procollagen 1 N-terminal propeptide; 25(OH)D: 25-hydroxyvitamin D; eGFR: glomerular filtration rate; BMD: bone mineral density; DKD: diabetic kidney disease; DR: diabetic retinopathy; DN: diabetic neuropathy.

**Table 3 biomolecules-13-00616-t003:** Correlation analysis of BMD with general conditions and biochemical indexes.

Index	Male				Female			
	Hip BMD		Lumbar Spine BMD		Hip BMD		Lumbar Spine BMD	
	*r*	*p*	*r*	*p*	*r*	*p*	*r*	*p*
Age (year)	−0.15	<0.05	0.07	0.08	−0.39	<0.05	−0.26	<0.05
BMI (kg/m^2^)	0.30	<0.05	0.22	<0.05	0.13	<0.05	0.21	<0.05
Diabetes duration (year)	−0.03	0.52	0.08	0.05	−0.08	0.05	−0.01	0.90
SBP (mmHg)	−0.02	0.64	−0.02	0.17	−0.03	0.56	0.04	0.40
DBP (mmHg)	0.02	0.58	−0.04	0.29	0.07	0.09	0.10	<0.05
HbA1c (%)	−0.01	0.78	0.01	0.89	0.05	0.34	0.03	0.18
FBG (mmol/L)	−0.02	0.64	−0.01	0.78	0.01	0.13	0.09	<0.05
TC (mmol/L)	−0.01	0.79	−0.08	0.06	−0.07	0.09	0.06	0.22
TG (mmol/L)	0.04	0.28	−0.01	0.89	0.03	0.52	0.09	<0.05
LDL-C (mmol/L)	−0.02	0.65	−0.10	<0.05	−0.10	<0.05	0.03	0.52
HDL-C (mmol/L)	0.34	<0.05	0.18	<0.05	0.48	<0.05	0.31	<0.05
UA (mmol/L)	0.07	0.09	0.04	0.33	0.01	0.92	0.18	<0.05
eGFR (mL/min/1.73^2^)	0.03	0.42	0.12	<0.05	0.24	<0.05	0.05	0.26
Ca (mmol/L)	−0.05	0.16	−0.01	0.95	−0.03	0.45	−0.04	0.37
PTH (pg/mL)	−0.09	<0.05	−0.05	0.23	−0.17	<0.05	−0.11	<0.05

Abbreviations: SBP: systolic blood pressure; DBP: diastolic blood pressure; BMI: body mass index; FBG: fasting blood glucose; HbA1c: glycosylated hemoglobin; UA: uric acid; Ca: calcium; TC: total cholesterol, TG: triglyceride; LDL-C: low-density lipoprotein cholesterol; HDL-C: high-density lipoprotein cholesterol; PTH: parathyroid hormone; eGFR: glomerular filtration rate; BMD: bone mineral density.

**Table 4 biomolecules-13-00616-t004:** Logistic regression analysis of blood lipid level and osteoporosis in different gendered patients.

Index	Crude OR	95% CI	*p*	Adjust OR	95% CI	*p*
Male TC (mmol/L)						
Low (≤3.5)	1			1		
Medium (3.5–5.2)	0.97	0.65, 1.46	0.90	0.80	0.49, 1.31	0.38
High (>5.2)	1.51	0.85, 2.66	0.16	1.22	0.62, 2.41	0.57
TG (mmol/L)						
Low (≤1.0)	1			1		
Medium (1.0–1.7)	0.97	0.61, 1.53	0.90	1.13	0.64, 1.99	0.68
High (>1.7)	0.83	0.51, 1.34	0.44	1.45	0.76, 2.80	0.26
LDL-C (mmol/L)						
Low (≤2.0)	1			1		
Medium (2.0–3.4)	1.45	0.96, 2.19	0.07	1.30	0.79, 2.15	0.30
High (>3.4)	2.46	1.39, 4.37	<0.05	1.79	0.90, 3.56	0.10
HDL-C (mmol/L)						
Low (≤1.0)	1			1		
Medium (1.0–1.5)	0.96	0.65, 1.41	0.83	0.83	0.52, 1.33	0.44
High (>1.5)	0.07	0.01, 0.53	<0.05	0.07	0.01, 0.53	<0.05
Female TC (mmol/L)						
Low (≤3.5)	1			1		
Medium (3.5–5.2)	1.27	0.82, 1.97	0.29	1.83	1.02, 3.27	<0.05
High (>5.2)	1.32	0.79, 2.22	0.29	2.13	1.07, 4.27	<0.05
TG (mmol/L)						
Low (≤1.0)	1			1		
Medium (1.0–1.7)	1.60	1.01, 2.54	<0.05	1.30	0.71, 2.39	0.39
High (>1.7)	1.06	0.68, 1.66	0.81	1.20	0.65, 2.22	0.55
LDL-C (mmol/L)						
Low (≤2.0)	1			1		
Medium (2.0–3.4)	1.47	0.99, 2.20	0.06	2.04	1.19, 3.50	<0.05
High (>3.4)	1.80	1.07, 3.05	<0.05	3.38	1.64, 6.98	<0.05
HDL-C (mmol/L)						
Low (≤1.0)	1			1		
Medium (1.0–1.5)	0.44	0.31, 0.64	<0.05	0.47	0.29, 0.76	<0.05
High (>1.5)	0.43	0.22, 0.85	<0.05	0.49	0.24, 0.96	<0.05

Abbreviations: TC: total cholesterol, TG: triglyceride; LDL-C: low-density lipoprotein cholesterol; HDL-C: high-density lipoprotein cholesterol.

## Data Availability

The data used to support the findings of this study are available from the corresponding authors upon request.

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
