# Peer review of "Correlation between Blood Lipid Level and Osteoporosis in Older Adults with Type 2 Diabetes Mellitus—A Retrospective Study Based on Inpatients in Beijing, China"

_biomolecules, 2023, doi:10.3390/biom13040616_

Round 1

Reviewer 1 Report

1)Abstract. Conclusion: In older T2DM patients, the effect of blood lipid levels is related to sex. HDL-C is a protective factor for OP in both men and women, while LDL-C independently predicts OP in postmenopausal women. Please, improve this paragraph and underline the novelty of the study.

2) 1. Introduction. L48-51. Cardiovascular disease and OP share risk factors, such as age, sedentary lifestyle, smoking, menopause, and deficiencies in vitamin D [7]. Genetics and physiological factors may contribute to these connections. Physiological factors include those influencing atherosclerosis and bone remodeling, such as lipid oxidation, osteoprotegerin, sclerosing  protein, and FGF-23. In order to discuss the previously reported points, important references are needed to be added on vitamin D deficiencies and bone metabolism in other systemic diseases, such as:

- Vitamin D deficiency and clinical correlations in systemic sclerosis patients: A retrospective analysis for possible future developments. PLoS One. 2017;12(6):e0179062. doi:10.1371/journal.pone.0179062

- Interactions between Muscle and Bone—Where Physics Meets Biology. Biomolecules 202010, 432. https://doi.org/10.3390/biom10030432

- Dickkopf-1 (Dkk-1) serum levels in systemic sclerosis and rheumatoid arthritis patients: correlation with the Trabecular Bone Score (TBS). Clin Rheumatol. 2018;37(11):3057-3062. doi:10.1007/s10067-018-4322-9

-  Pathological Crosstalk between Metastatic Breast Cancer Cells and the Bone Microenvironment. Biomolecules 202010, 337. https://doi.org/10.3390/biom10020337

3) 1. Introduction. L 56-58. Here, we retrospectively analyzed the relationships between various blood lipid components and OP in older T2DM patients, so as to add a new direction for the discovery of OP risk factors in these patients and to provide evidence for preventing OP and  fractures in such patients. Please improve this paragraph and underline the novelty of the study.

4) 3. Results L108-116. 3.1. Patient Features and Measurements in the Different Groups  The OP group was characterized by having more female members of older age and  lower BMI, differing significantly from normal and osteopenia patients (P < 0.05, respectively). GFR was also reduced in the OP group (P<0.05) and highest in normal 112 (P><0.05), while the UA was significantly lower in OP patients and highest in normal BMD 113 patients (p><0.05). Furthermore, LDL-C concentrations were markedly higher in both OP 114 and osteopenia patients while HDL-C was markedly higher in the normal group (all P ><  0.05). The levels of β-CTx, OC, and P1NP were lowest in the normal group and highest in  the OP group (p<0.05, respectively) while normal patients had higher levels of 25 (OH) D, 117 in contrast to OP patients, where the levels were significantly lower (all P >< 0.05) (shown  in Table 1). Please insert the exact statistically significant p-values

5) Table 3. Logistic regression analysis of lipid and OP and other tables. Please insert the exact statistically significant p-values.

6) 4. Discussion L168-170. Although there has been much investigation into an association between T2 DM and  OP, the issue remains controversial. Even T2DM patients with normal or even high BMD have an increased risk of fracture [10]. Please, summarise here the most important results of the study.

7) 5. Conclusion L270-274. To sum up, this clinical study showed that age, female sex, and high LDL-C levels  can increase the likelihood of OP in T2DM patients, while high BMI and high HDL-C  levels are protective against OP. Thus, clinicians should be aware of the importance of  blood lipid levels and take necessary interventional measures for their control, especially  in the case of LDL-C, in T2DM patients. Please, improve this paragraph and underline the clinical implications of the study.

Author Response

ear Professor,

Thank you very much for your decision letter and advice on our manuscript entitled “Correlation between Blood Lipid level and Osteoporosis in Older Adults with Type 2 Diabetes Mellitus-A Retrospective Study Based on Inpatients in Beijing, China”. We also thank the reviewer for the constructive and positive comments and suggestions. Accordingly, we have revised the manuscript. All amendments are highlighted in red in the revised manuscript. In addition, point-by-point responses to the comments are listed below this letter.

We hope that the revision is acceptable for the publication in your journal.

Look forward to hearing from you soon.

With best wishes,

Yours sincerely,

Xin Zhao

Xiaomei Zhang

Reviewer 2 Report

The research work is of great interest. 

By the way, here are a number of technical comments:

-In the abstract, you need to decipher the eGFR – 22 line. 

-52 line – you need to decipher FGF-23. 

-67 line – you need to remove gestational diabetes, since menopausal women are included. 

-88 line – you need to specify how C-LDL was measured. 

-In 2.5. Statistical Analysis, you need to specify that the logistic regression method was used.

 There are also two questions to be mentioned:

 Question 1. In Table 3 and 4, please specify, by what principle are lipids divided into three groups?

 Suggestions on the essence of the work. 

 The analysis of risk factors for osteoporosis separately in men and women with DM2 is of the greatest interest. It is more logical to present a list of general characteristics, biochemical indices, BMD and bone metabolism markers among the three groups in different gender patients. I suggest that instead of Table 1, two tables should be presented separately for men and women, same for table 2. 

I recommend do not represent Figure 1 and Table 3 in the state.

 Question 2. How many had a history of fractures in each group (among men and women).

Author Response

Dear Professor,

Thank you very much for your decision letter and advice on our manuscript entitled “Correlation between Blood Lipid level and Osteoporosis in Older Adults with Type 2 Diabetes Mellitus-A Retrospective Study Based on Inpatients in Beijing, China”. We also thank the reviewer for the constructive and positive comments and suggestions. Accordingly, we have revised the manuscript. All amendments are highlighted in red in the revised manuscript. In addition, point-by-point responses to the comments are listed below this letter.

We hope that the revision is acceptable for the publication in your journal.

Look forward to hearing from you soon.

With best wishes,

Yours sincerely,

Xin Zhao

Xiaomei Zhang

Reviewer 3 Report

Here, Zhao and colleagues, analyze a cohort of 1158 diabetic patients with or without osteoporosis. The authors showed that high levels of LDL-C represent an independent risk factor for the osteoporotic condition, while HDL-C seems to be protective.

The topic of this study is interesting, and the large cohort supports the authors' conclusion.

However, I have some major issues:

-        Is there any association between the presence of osteoporosis and diabetes control such as HbA1c, diabetes complications and diabetes treatments? You could include these variables in the logistic regression.

-        -Other studies have previously analyzed the correlation between osteoporosis and blood lipids in type 2 diabetic patients. Please, update the bibliography (e.g., 10.4158/EP.13.6.620; 10.2147/DMSO.S372348).

 Minor comments:

-        Specify the actual value of the p-value.

-        Replace “normal group” with “non-osteoporotic group”.

-        In figure 1, I suggest showing only the significant correlations.

Author Response

(The authors gave the same response as above.)

Reviewer 4 Report

In this retrospective study, authors compared blood lipids, BMD, and bone metabolism markers among three groups of older T2DM patients. Authors found higher LDL-C as an independent risk factor for osteoporosis in postmenopausal women and higher HDL-C as a protective factor for osteoporosis in both men and women.

The paper is well structured.

Unfortunately, the article has several limitations.

1/ The title of the article is misleading. The authors did not compare blood lipid metabolism and osteoporosis but only compared some blood lipid components between the control group and the groups of patients with osteopenia and osteoporosis.

2/Criteria for classification into the control group, osteopenia, and osteoporosis are missing.

3/ The introduction needs to expand much more to specify the pathways leading to the development of osteoporosis.

4/ Unfortunately, I miss the originality of the study design.

Author Response

(The authors gave the same response as above.)

Round 2

Reviewer 3 Report

The authors addressed all my concerns